# Efficient Access to 3,5-Disubstituted 7-(Trifluoromethyl)pyrazolo[1,5-*a*]pyrimidines Involving S*_N_*Ar and Suzuki Cross-Coupling Reactions

**DOI:** 10.3390/molecules25092062

**Published:** 2020-04-28

**Authors:** Badr Jismy, Abdellatif Tikad, Mohamed Akssira, Gérald Guillaumet, Mohamed Abarbri

**Affiliations:** 1Laboratoire de Physico-Chimie des Matériaux et des Electrolytes pour l’Energie (PCM2E), EA 6299, Avenue Monge, Faculté des Sciences, Université de Tours, Parc de Grandmont, 37200 Tours, France; 2Laboratoire de Chimie Moléculaire et Substances Naturelles, Faculté des Sciences, Université Moulay Ismail, B.P. 11201, Zitoune, Meknès 50050, Morocco; 3Laboratoire de Chimie Physique & de Chimie Bioorganique, URAC 22, Université Hassan II de Casablanca, B.P. 146, Mohammedia 28800, Morocco; 4Institut de Chimie Organique et Analytique (ICOA), Université d’Orléans, UMR CNRS 7311, BP 6759, Rue de Chartres, CEDEX 2, 45067 Orléans, France

**Keywords:** 3-amino-1*H*-pyrazole, ethyl 4,4,4-trifluoro-2-butynoate, 3-bromo-7-(trifluoromethyl)pyrazolo[1,5-*a*]pyrimidin-5-one, C–O bond activation, Suzuki–Miyaura cross-coupling

## Abstract

An efficient and original synthesis of various 3,5-disubstituted 7-(trifluoromethyl)pyrazolo[1,5-*a*]pyrimidines is reported. A library of compounds diversely substituted in C-3 and C-5 positions was easily prepared from a common starting material, 3-bromo-7-(trifluoromethyl)pyrazolo[1,5-*a*]pyrimidin-5-one. In C-5 position, a S*_N_*Ar type reaction was achieved by first activating the C–O bond of the lactam function with PyBroP (Bromotripyrrolidinophosphonium hexafluorophosphate), followed by the addition of amine or thiol giving monosubstituted derivatives, whereas in C-3 position, arylation was performed via Suzuki–Miyaura cross-coupling using the commercially available aromatic and heteroaromatic boronic acids. Moreover, trifluoromethylated analogues of potent Pim1 kinase inhibitors were designed following our concise synthetic methodology.

## 1. Introduction

Pyrimidine fused heterocycles constitute an important class of heterocyclic compounds, having various applications in many fields owing to their chemical and biological properties. In particular, pyrazolo[1,5-*a*]pyrimidines are one of the most interesting pyrimidine derivatives displaying a wide range of pharmacological activities, as antimicrobial [1,2], antitumor [3], anticancer [4,5,6,7,8], anxiolytic [9,10], antidepressant [11,12,13], and antiepileptic agents [14]. Moreover, pyrazolo[1,5-*a*]pyrimidines are also known as antagonists of serotonin 5-HT6 receptors [15], and also act as inhibitors of histone lysine demethylase 4D (KDM4D) [16] as well as several kinases such as Pim kinase [17,18], threonine tyrosine kinase (TTK) [19], and cyclin-dependent kinases (CDKs) [20]. In addition, some pyrazolo[1,5-*a*]pyrimidine derivatives were found to have an antiviral effect against the respiratory syncytial virus (RSV) [21] and hepatitis C virus [22], and are inhibitors of phosphodiesterase (PDE) 2A [23] and 10A [24,25,26], promising new targets for the treatment of cognitive disorders and schizophrenia, respectively.

The incorporation of a fluorine atom into potential pharmaceutical substances has been successfully used as a key strategy for enhancing the activity of drugs or drug candidates [27,28]. The introduction of fluorine atoms leads to a change in the solubility and lipophilicity of compounds, thus increasing their bioavailability and their cell membranes permeability. As a consequence, the combination between the pyrazolo[1,5-*a*]pyrimidine scaffold and the interesting properties of the fluorine atom could give rise to new heterocycles displaying potent biological activities. Synthetically, although many approaches have been reported for the synthesis of non-fluorinated pyrazolo[1,5-*a*]pyrimidines, the incorporation of fluorinated groups in this heterocycle remains limited [29,30,31,32,33,34,35,36]. Recently, we described the condensation of 5-substituted 3-aminopyrazoles with ethyl 4,4,4-trifluoro-2-butynoate, affording regioselectively the 2-substituted 7-(trifluoromethyl)pyrazolo[1,5-*a*]pyrimidin-5-ones in fair to good yields [37]. Continuing our research directed towards the development of new heterocyclic compounds containing the fluorine atoms [38,39,40,41], we report herein an efficient approach for synthesizing 3,5-disubstituted 7-(trifluoromethyl)pyrazolo[1,5-*a*]pyrimidines in good yields through S*_N_*Ar and Suzuki–Miyaura cross-coupling reactions, using 3-bromo-7-(trifluoromethyl)pyrazolo[1,5-*a*]pyrimidin-5-one as a starting material.

## 2. Results and Discussion

Our synthesis starts with the generation of 7-(trifluoromethyl)pyrazolo[1,5-*a*]pyrimidin-5-one core **3** following our reported procedure [41]. The reaction between 3-aminopyrazole **1** and ethyl 4,4,4-trifluoro-2-butynoate **2** in 1,4-dioxane at 110 °C under microwave irradiation for 2 h, followed by the addition of NaOMe (2 equiv.) and stirring for 12 h at room temperature, gave the only regioisomer **3** in 63% yield (Scheme 1). Selective bromination of **3** with *N*-bromosuccinimide in dichloromethane afforded the bromo derivative **4** in 94% yield. The 3-bromo-7-(trifluoromethyl)pyrazolo[1,5-*a*]pyrimidin-5-one **4** served as a building block for the synthesis of 3,5-disubstituted 7-(trifluoromethyl)pyrazolo[1,5-*a*]pyrimidines having two new points of diversity.

Recently, we reported one pot S*_N_*Ar reaction strategy to functionalize the pyrazolo[1,5-*a*]pyrimidine moiety at C-5 position through C-O activation of the lactam function with PyBroP (bromotripyrrolidinophosphonium hexafluorophosphate) as the activating reagent [41]. We extended this approach to the brominated compound **4** in order to substitute the pyrazolo[1,5-*a*]pyrimidine ring differently at C-3 and C-5 positions. The C–O bond activation was carried out with PyBroP (Bromotripyrrolidinophosphonium hexafluorophosphate) (1.3 equiv.) in the presence of Et_3_N (3 equiv.) in 1,4-dioxane at room temperature for 2 h. Then, the addition of various amines (1.5 equiv.) in 1,4-dioxane at 110 °C for 12 h led to the desired 5-amino-7-(trifluoromethyl)pyrazolo[1,5-*a*]pyrimidines **5a**–**f** in 74–98% yields (Scheme 2). A one-pot reaction using benzylamine and 4-methoxybenzylamine as the nucleophilic reagents provided the corresponding trifluoromethylated 5-aminopyrazolo[1,5-*a*]pyrimidines **5a** and **5b** in the same 90% yield. Under the same conditions, aliphatic amines bearing protected or unprotected functions such as NHBoc, free hydroxyl group and triple bond, reacted well and afforded the expected heterocycles **5c**–**e** in 74%, 84%, and 96% yields, respectively (Scheme 2). When morpholine was used as a secondary amine, this S*_N_*Ar reaction was also efficiently performed and the compound **5f** was isolated in 98%. In view of the success of amine addition, this reaction was extended to sulfur compounds such as *p*-methoxythiophenol (PMPSH) giving successfully the desired product **5g** in 73% yield (Scheme 2).

In order to prepare a library of new 3,5-disubstituted 7-(trifluoromethyl) pyrazolo[1,5-*a*]pyrimidines, bromine in C-3 position of **5a–f** was subjected to a Suzuki–Miyaura cross-coupling. To optimize the coupling conditions, the reaction between compound **5f** and 4-methoxyphenylboronic acid was studied as a model (Table 1). First, this coupling was carried out with *p*-methoxyphenylboronic acid (2 equiv.) in the presence of PdCl_2_(PPh_3_)_2_ (10 mol%) as the catalyst and sodium carbonate (2 equiv.) as the base in a mixture of dioxane/water (4/1) at 110 °C. Under these conditions, the starting material was completely consumed after 12 h, and the analysis of the crude ^1^H NMR spectrum (see Appendix A) showed 10% of the coupled product **6a** along with 90% of debrominated compound **7** as a byproduct (Table 1, entry 1). As a proper combination between catalyst, base, and solvent is very important for the success of the reaction, the influence of these parameters was then examined. Using PdCl_2_dppf instead of PdCl_2_(PPh_3_)_2_ improved slightly the **6a**/**7** ratio from (10:90) to (20:80) (Table 1, entries 1–2). Under the same conditions, replacing Na_2_CO_3_ by K_2_CO_3_ resulted in a similar **6a**/**7** ratio (Table 1, entry 3). The replacement of PdCl_2_dppf with XPhosPdG2/XPhos tandem at 135 °C affected slightly the formation of byproduct **7** since its yield decreased to 70% and the yield of the desired product **6a** increased to 30% (Table 1, entry 4). To reduce the reaction time and to improve the yield of **6a** at the expense of **7**, conventional thermal heating was replaced by microwave irradiation (MW) to promote the reaction. This heating mode seems beneficial for the reaction since the time was reduced to only 40 min and the percentage of **6a** in the mixture increased to 45% (Table 1, entry 5). Gratifyingly, when adding ethanol as an organic solvent instead of dioxane, the starting material was fully converted to the desired compound **6a** which was obtained in excellent 93% isolated yield, after purification by silica-gel column chromatography (Table 1, entry 6). A significant effect of the base on the formation of the undesired product **7** was observed when Na_2_CO_3_ was used instead of K_2_CO_3_ (Table 1, entry 7). When the reaction was conducted without ligand, both products **6a** and **7** were formed equally, indicating that the Xphos ligand plays a critical role in avoiding the formation of byproduct **7** (Table 1, entry 8). By reducing the amounts of catalyst and ligand to 5 mol%, a decrease in the yield of coupling product **6a** was observed (Table 1, entry 9).

On analyzing the synthesis of compound **6a**, which involved a C–O bond activation reaction using morpholine as amine at C-5 followed by the Suzuki–Miyaura cross-coupling at C-3 (Scheme 3, Path A), the overall yield was found to be 91% over two steps. This sequence could be reversed by conducting the Suzuki–Miyaura coupling first at C-3 followed by S*_N_*Ar reaction at C-5 (Scheme 3, Path B). Following the synthetic route B, the overall yield of **6a** decreased to 77% and some problems were encountered during the purification of the intermediate **8** because of the presence of unprotected lactam function (Scheme 3, Path B). Therefore, the efficient, convergent, and interesting synthetic pathway A was applied to generate several new 3,5-disubstituted 7-(trifluoromethyl)pyrazolo[1,5-*a*]pyrimidines. Encouraged by the good yield of **6a** obtained in two sequential steps (path A), this synthesis was attempted according to a one-pot strategy. In a sealed vial and reproducing the same conditions as those of the first and second steps (Path A), without isolating intermediate **5f**, the desired product **6a** was not observed since only 61% of debrominated by-product **7** was isolated. This debromination was likely induced by the presence of triethylamine and morpholine in the reaction medium.

After establishing the best conditions to carry out the Suzuki coupling at C-3 of **5f**, we extended this coupling to monosubstituted substrates **5a**–**e** using commercially available aromatic and heteroaromatic boronic acids. In all cases, direct arylation at C-3 produced the corresponding 3,5-bifunctionnalized 7-(trifluoromethyl)pyrazolo[1,5-*a*]pyrimidines **6a**–**m** in good to excellent isolated yields whatever the nature of the amine at C-5. The results are summarized in Scheme 4.

As illustrated in Scheme 4, unsubstituted aryl such as phenyl, biphenyl, and naphthyl were successfully coupled under optimized conditions, and the desired products **6b**, **6d**, **6f**, **6k**, and **6m** were isolated in yields ranging from 74% and 91%. A phenylboronic acid bearing a methoxy group, as an electron-donating group, at the *para* position was easily coupled with **5f**, giving the desired compound **6a** in 93% yield. In addition, a free alcohol moiety was tolerated at the *para* position of phenylboronic acid providing compound **6g** in 82% yield. A phenylboronic acid with CF_3_, as an electron-withdrawing group, was also tolerated, yielding the expected product **6l** in 77% yield (Scheme 4).

After the good results obtained with different aryls, this coupling was extended to heteroaryl boronic acids. Interestingly, using 2-thienyl as an heteroarylboronic acid in coupling reaction with 7-(trifluoromethyl)pyrazolo[1,5-*a*]pyrimidines containing morpholino or benzylamino group at C-5 position, worked well and the desired heterocycles were isolated in good yields (**6c** (79%) and **6e** (76%)). Similarly, the cross-coupling between 3-pyridinylboronic acid or 4-dibenzothienylboronic acid and **5a**, provided the coupled products in excellent yields (**6i** (91%) and **6j** (92%)). However, the isolated yield obtained with the coupling of 1*H*-pyrazole-4-boronic acid and **5a** did not exceed 50% which could be explained by the presence of a free NH-pyrazole bond (Scheme 4).

To demonstrate the synthetic potential of our method, we turned our attention to the synthesis of biologically active compounds. With the introduction of a suitable aryl at C-3 and amine group at C-5 of the pyrazolo[1,5-*a*]pyrimidine core, we were able to design the fluorinated pyrazolo[1,5-*a*]pyrimidines **6k** and **6l**, analogues of compounds **10** and **11**, which have a nanomolar activities against Pim1 kinase (**10**, IC_50_ = 18 nM) and (**11**, IC_50_ = 27 nM), respectively (Scheme 5) [17,42]. In order to prepare the trifluoromethylated analogue of the potent Pim1 kinase inhibitor **12** (IC_50_ = 1.5 nM), the Boc protected amine **6m** was treated with an excess of trifluoroacetic acid at room temperature, yielding the desired diamine **9** in 95% (Scheme 5). It is noteworthy that there is a correlation between prostate cancer disease and Pim-1 since the DNA microarray analyses showed that this kinase was overexpressed in human prostate cancer [43].

## 3. Materials and Methods

### 3.1. General Methods

All reagents were purchased from commercial suppliers and were used without further purification. Triethylamine and DCM were distilled over calcium hydride. 1,4-Dioxane was distilled over sodium and benzophenone. Reactions were monitored by thin-layer chromatography (TLC) analysis using silica gel (60 F254) plates. The compounds were visualized by longwave (365 nm) or shortwave (254 nm) UV light. Column chromatography was performed using silica gel 60 (230–400 mesh, 0.040–0.063 mm) to purify the product.

All ^1^H NMR, ^13^C NMR, and ^19^F NMR spectra were recorded with a 300 MHz Bruker Avance FT-NMR spectrometer (300 MHz, 75 MHz, or 282 MHz, respectively, (Billerica, MA, USA).

All chemical shifts are given in ppm and they are referenced to tetramethylsilane (TMS) as an internal standard. ^1^H NMR assignment abbreviations are indicated as follows: singlet (s), doublet (d), triplet (t), quartet (q), broad singlet (br s), doublet of doublets (dd), triplet of doublets (td), doublet of a triplets (dt), and multiplet (m). Coupling constants (J) are reported in Hertz (Hz). Electrospray ionization high-resolution mass spectrometry experiments were performed with a hybrid tandem quadrupole/time-of flight (Q-TOF) instrument, equipped with a pneumatically assisted electrospray (Zspray) ion source (Micromass, Manchester, UK) operated in positive mode. Melting points were measured using banc Kofler.

#### Microwave Assisted Reactions

The microwave-assisted reactions were performed using a Monowave 300 (microwave synthesis reactor: Anton Paar, 300 W maximum power, Graz, Austria). Microwave irradiation was carried out in sealed 10–30 mL vessels (borosilicate glass) with a PTFE (polytetrafluoroethylene)-coated silicon septum and closed with a snap cap made of PEEK (polyetheretherketone). The temperatures were measured on the surface of the vial with an IR sensor (measuring range: 30 to 300 °C; uncertainty: ±5 °C) and were measured with high precision in the reaction mixture with a ruby thermometer (measuring range: 30 to 300 °C; uncertainty: ±2 °C) that could be read directly from the instrument screen. The reaction time was measured from when the reaction mixture reached the stated temperature for temperature controlled experiments. The pressure was measured with a non-invasive pressure sensor located in the swiveling cover of the Monowave 300 (measuring range: 0 to 30 bar; uncertainty: ±1.5 bar).

### 3.2. Synthesis of 3-Bromo-7-(trifluoromethyl)pyrazolo[1,5-a]pyrimidin-5-one 4: Bromination of 3-bromo-7-(trifluoromethyl)pyrazolo[1,5-a]pyrimdin-5-one 3

NBS (1.05 equiv.) was added to a solution of pyrazolo[1,5-*a*]pyrimdin-5-one **3** (1 equiv.) in 8 mL of CH_2_Cl_2_ and the reaction was stirred under an argon atmosphere at room temperature for 12 h. The progress of the reaction was monitored by TLC (PE/EtOAc, 7/3). After completion of the reaction, evaporation of the solvent under reduced pressure provided the crude product, which was purified by column chromatography (PE/EtOAc, 7.5/2.5) to afford the final product **4** as a white solid in 94% yield; melting point: 209–211 °C. ^1^H NMR (300 MHz, DMSO-*d*_6_): *δ* 13.00 (s, 1H), 8.07 (s, 1H), 6.74 (s, 1H). ^13^C NMR (75 MHz, DMSO-*d*_6_): *δ* 159.7, 144.8, 140.5, 135.7 (q, *J* = 36.8 Hz), 119.2 (q, *J* = 274.6 Hz), 108.0, 75.1. ^19^F NMR (282 MHz, DMSO-*d*_6_): *δ* −66.65. HRMS (ESI) *m*/*z* [M + H]^+^ calcd for C_7_H_4_BrF_3_N_3_O: 281.9484; found: 281.9474. 

### 3.3. General Procedure for the Synthesis of 5-Amino (or Mercapto) 3-bromo-7-(trifluoromethyl)pyrazolo[1,5-a]pyrimidines 5a–g by the Direct Amination or Thiolation via C-OH Bond Activation with PyBroP

In a sealed tube, PyBroP (1.44 mmol, 621 mg, 1.3 equiv.) and Et_3_N (3.33 mmol, 0.46 mL, 3 equiv.) were added successively to a solution of **4** (1.11 mmol, 150 mg, 1 equiv.) in 1.4-dioxane (4 mL). The mixture was degassed by argon bubbling and then stirred at room temperature for 2 h. The corresponding amine or thiol (1.5 equiv.) was then added, and the mixture was stirred for 12 h at 110 °C. After cooling, the solvents were evaporated under reduced pressure, and the crude residue was purified by silica gel column chromatography to give desired 5-aminated (thiolated) 3-bromo-7-(trifluoromethyl)pyrazolo[1,5-*a*]pyrimdines **5a**–**g**.

*3-Bromo-5-[N-(4-methoxybenzyl)amino]-7-(trifluoromethyl)pyrazolo[1,5-a]pyrimidine* (**5a**). The purification of the crude product by chromatography on silica gel is carried out using (PE/EtOAc: 9/1) to afford **5a** as a yellow solid in 90% yield. m.p. 149–151 °C; ^1^H NMR (300 MHz, CDCl_3_): *δ* 7.97 (s, 1H), 7.35 (d, *J* = 8.7 Hz, 2H), 6.92 (d, *J* = 8.7 Hz, 2H), 6.44 (s, 1H), 5.36 (br s, 1H), 4.66 (q, *J* = 5.3 Hz, 2H), 3.83 (s, 3H); ^13^C NMR (75 MHz, Acétone-*d*_6_): *δ* 159.2, 154.9, 146.5, 144.4, 133.9 (q, *J* = 36.5 Hz), 130.3, 129.5 (2C), 119.4 (q, *J* = 273.3 Hz), 113.8 (2C), 100.9, 79.3, 54.6, 44.3; ^19^F NMR (282 MHz, CDCl_3_): *δ* −68.47; HRMS (ESI) *m*/*z* [M + H]^+^ calcd for C_15_H_13_BrF_3_N_4_O: 401.0219; found: 104.0224.

*5-(N-Benzylamino)-3-bromo-7-(trifluoromethyl)pyrazolo[1,5-a]pyrimidine* (**5b**). The purification of the crude product by chromatography on silica gel is carried out using (PE/EtOAc: 9/1) to afford **5b** as a yellow solid in 90% yield. m.p. 147–149 °C; ^1^H NMR (300 MHz, CDCl_3_): *δ* 7.97 (s, 1H), 7.42–7.35 (m, 5H), 6.46 (s, 1H), 5.43 (br s, 1H), 4.75 (d, *J* = 5.3 Hz, 2 H); ^13^C NMR (75 MHz, Acetone-*d*_6_): *δ* 155.0, 146.5, 144.4, 138.5, 133.9 (q, *J* = 36.8 Hz), 128.4, 128.1, 127.3, 119.4 (q, *J* = 273.3 Hz), 100.8, 79.3, 44.7. ^19^F NMR (282 MHz, CDCl_3_): *δ* −68.46; HRMS (ESI) *m*/*z* [M + H]^+^ calcd for C_14_H_11_BrF_3_N_4_: 371.0114; found: 371.0119.

*3-Bromo-5-[2-(tert-butoxycarboxylamino)-N-ethylamino]-7-(trifluoromethyl)pyrazolo[1,5-a]pyrimidine* (**5c**). The purification of the crude product by chromatography on silica gel is carried out using (PE/EtOAc: 8/2) to afford **5c** as a yellow solid in 74% yield. m.p. 155–157 °C; ^1^H NMR (300 MHz, CDCl_3_): *δ* 7.93 (s, 1H), 6.48 (s, 1H), 6.43 (br s, 1H), 5.12 (br s, 1H), 3.73–3.63 (m, 2H), 3.51–3.41 (m, 2H), 1.45 (s, 9H); ^13^C NMR (75 MHz, Acetone-*d*_6_): *δ* 156.2, 165.4, 146.5, 144.4, 133.8 (q, *J* = 37.4 Hz), 119.4 (q, *J* = 273.3 Hz), 101.0, 79.2, 78.0, 41.5, 39.4, 27.7 (3C); ^19^F NMR (282 MHz, CDCl_3_): *δ* −68.41; HRMS (ESI) *m*/*z* [M + H]^+^ calcd for C_14_H_18_BrF_3_N_5_O_2_: 424.0590; found: 424.0593.

*3-Bromo-5-N-[4-(hydroxycyclohexyl)amino]-7-(trifluoromethyl)pyrazolo[1,5-a]pyrimidine* (**5d**). The purification of the crude product by chromatography on silica gel is carried out using (PE/EtOAc: 6/4) to afford **5d** as a yellow solid in 84% yield. m.p. 195–197 °C; ^1^H NMR (300 MHz, Acetone-*d*_6_): *δ* 7.93 (s, 1H), 7.21 (d, *J* = 6.32 Hz, 1H), 4.07–4.03 (m, 1H), 3.82 (d, *J* = 4.5 Hz, 1H), 3.63–3.57 (m, 1H), 2.96–2.93 (m, 2H), 2.10–2.05 (m, 2H), 1.46–1.37 (m, 4H); ^13^C NMR (75 MHz, Acetone-*d*_6_): *δ* 154.5, 146.6, 144.4, 133.7 (q, *J* = 37.1 Hz), 119.4 (q, *J* = 273.3 Hz), 101.0, 79.0, 68.8, 49.3, 33.8 (2C), 29.8 (2C); ^19^F NMR (282 MHz, Acetone-*d*_6_): *δ* −68.97; HRMS (ESI) *m*/*z* [M + H]^+^ calcd for C_13_H_15_BrF_3_N_4_O: 379.0376; found: 379.0378.

*3-Bromo-N-[(prop-2-ynyl)amino]-7-(trifluoromethyl)pyrazolo[1,5-a]pyrimidine* (**5e**). The purification of the crude product by chromatography on silica gel is carried out using (PE/EtOAc: 8.5/1.5) to afford **5e** as a yellow solid in 96% yield. m.p. 149–151 °C; ^1^H NMR (300 MHz, CDCl_3_): *δ* 7.99 (s, 1H), 6.51 (s, 1H), 5.33 (br s, 1H), 4.39 (dd, *J* = 5.2, 2.5 Hz, 2H), 2.34 (t, *J* = 2.5 Hz, 1H); ^13^C NMR (75 MHz, Acetone-*d*_6_): *δ* 154.5, 146.2, 144.6, 134.1 (q, *J* = 36.9 Hz), 119.4 (q, *J* = 273.4 Hz), 100.5, 79.7, 79.6, 72.1, 30.2; ^19^F NMR (282 MHz, CDCl_3_): *δ* −68.49; HRMS (ESI) *m*/*z* [M + H]^+^ calcd for C_10_H_7_BrF_3_N_4_: 318.9801; found: 318.9802.

*3-Bromo-5-morpholino-7-(trifluoromethyl)pyrazolo[1,5-a]pyrimidine* (**5f**). The purification of the crude product by chromatography on silica gel is carried out using (PE/EtOAc: 8/2) to afford **5f** as a yellow solid in 98% yield. m.p. 182–184 °C; ^1^H NMR (300 MHz, CDCl_3_): *δ* 7.99 (s, 1 H), 6.74 (s, 1H), 3.86–3.79 (m, 8H); ^13^C NMR (75 MHz, Acetone-*d*_6_): *δ* 155.0, 146.0, 144.9, 134.5 (q, *J* = 36.7 Hz), 119.6 (q, *J* = 273.8 Hz), 97.4 (q, *J* = 4.6 Hz), 79.1, 66.1 (2C), 45.2 (2C); ^19^F NMR (282 MHz, CDCl_3_): *δ* −68.43; HRMS (ESI) *m*/*z* [M + H]^+^ calcd for C_11_H_11_BrF_3_N_4_O: 351.0063; found: 351.0063.

*3-Bromo-5-((4-methoxyphenyl)thio)-7-(trifluoromethyl)pyrazolo[1,5-a]pyrimidine* (**5g**). The purification of the crude product by chromatography on silica gel is carried out using (PE/EtOAc: 9.5/0.5) to afford **5g** as a yellow solid in 73% yield. m.p. 135–137 °C; ^1^H NMR (300 MHz, CDCl_3_): *δ* 8.12 (s, 1H), 7.58 (d, *J* = 8.8 Hz, 2H), 7.05 (d, *J* = 8.8 Hz, 2H), 6.73 (s, 1H), 3.91 (s, 3H); ^13^C NMR (75 MHz, Acetone-*d*_6_): *δ* 163.9, 161.7, 145.8, 145.5, 137.2 (2C), 133.4 (q, *J* = 37.5 Hz), 119.3 (q, *J* = 274.1 Hz), 117.7, 115.5 (2C), 105.6 (q, *J* = 4.4 Hz), 83.6, 55.1; ^19^F NMR (282 MHz, CDCl_3_): *δ* −68.59; HRMS (ESI) *m*/*z* [M + H]^+^ calcd for C_14_H_10_BrF_3_N_3_OS: 403.9675; found: 403.9678.

### 3.4. General Procedure for C-3 Suzuki–Miyaura Cross-Coupling: Synthesis of 3,5-Disubstituted 7-(trifluoromethyl)pyrazolo[1,5-a]pyrimdines ***6a**–**m***

A mixture of 5-aminated 3-bromo-7-(trifluoromethyl)pyrazolo[1,5-*a*]pyrimdines **5a**–**g** (1 equiv.), boronic acid (2 equiv.), and K_2_CO_3_ (2 equiv.) in EtOH (4 mL) and water (1 mL) was thoroughly degassed with a stream of argon. Then, XPhos (10 mol%) and XPhosPdG2 (10 mol%) were added and the microwave vial containing the mixture was capped and inserted into microwave reactor. The reaction mixture was irradiated at 135 °C for 40 min. After that, the mixture was filtered through a Celite, washed with EtOAc, and concentrated under reduced pressure. Purification of the crude via column chromatography afforded desired 3,5-disubstituted 7-(trifluoromethyl)pyrazolo[1,5-*a*]pyrimidines **6a**–**m**, which was characterized by ^1^H, ^13^C NMR, ^19^F NMR and HRMS.

*3-(4-Methoxyphenyl)-5-morpholine-7-(trifluoromethyl)pyrazolo[1,5-a]pyrimidine* (**6a**). The purification of the crude product by chromatography on silica gel is carried out using (PE/EtOAc: 8/2) to afford **6a** as a yellow solid in 93% yield. m.p. 243–245 °C; ^1^H NMR (300 MHz, CDCl_3_): *δ* 8.31 (s, 1H), 7.90 (d, *J* = 8.8 Hz, 2H), 7.00 (d, *J* = 8.8 Hz, 2H), 6.74 (s, 1H), 3.90–3.88 (m, 4H), 3.87 (s, 3H), 3.79–3.76 (m, 4H); ^13^C NMR (75 MHz, CDCl_3_): *δ* 157.85, 154.1, 143.7, 135.2 (q, *J* = 36.1 Hz), 126.9 (2C), 124.9, 120.5, 119.5 (q, *J* = 274.5 Hz), 114.2 (2C), 107.0, 95.0 (q, *J* = 4.5 Hz), 66.45 (2C), 55.3, 45.3 (2C); ^19^F NMR (282 MHz, CDCl_3_): *δ* −68.74. HRMS (ESI) *m*/*z* [M + H]^+^ calcd for C_18_H_18_F_3_N_4_O_2_: 379.1376; found: 379.1374.

*5-Morpholino-3-phenyl-7-(trifluoromethyl)pyrazolo[1,5-a]pyrimidine* (**6b**). The purification of the crude product by chromatography on silica gel is carried out using (PE/EtOAc: 8/2) to afford **6b** as a yellow solid in 80% yield. m.p. 230–232 °C; ^1^H NMR (300 MHz, CDCl_3_): *δ* 8.37 (s, 1H), 7.98 (dd, *J* = 8.4, 1.2 Hz, 2H), 7.45 (t, *J* = 7.7 Hz, 2H), 7.25 (tt, *J* = 7.7, 1.2 Hz, 1H), 6.76 (s, 1H), 3.91–3.87 (m, 4H), 3.81–378 (m, 4H); ^13^C NMR (75 MHz, CDCl_3_): *δ* 154.3, 145.35, 144.0, 135.3 (q, *J* = 36.6 Hz), 132.3, 128.7 (2C), 125.8, 125.6 (2C), 119.5 (q, *J* = 274.6 Hz), 107.0, 95.1 (q, *J* = 4.5 Hz), 66.4 (2 C), 45.25 (2C); ^19^F NMR (282 MHz, CDCl_3_): *δ* −68.67; HRMS (ESI) *m*/*z* [M + H]^+^ calcd for C_17_H_16_F_3_N_4_O: 349.1271; found: 349.12680.

*5-Morpholino-3-(thien-2-yl)-7-(trifluoromethyl)pyrazolo[1,5-a]pyrimidine* (**6c**). The purification of the crude product by chromatography on silica gel is carried out using (PE/EtOAc: 8/2) to afford **6c** as a yellow solid in 79% yield. m.p. 233–235 °C; ^1^H NMR (300 MHz, CDCl_3_): *δ* 8.27 (s, 1H), 7.42 (dd, *J* = 3.5, 1.1 Hz, 1H), 7.23 (dd, *J* = 5.1, 1.1 Hz, 1 H), 7.10 (dd, *J* = 5.1, 3.5 Hz, 1H), 6.74 (s, 1H), 3.91–3.87 (m, 4H), 3.83–3.80 (m, 4H). ^13^C NMR (75 MHz, CDCl_3_): *δ* 154.3, 144.6, 143.2, 135.3 (q, *J* = 36.9 Hz), 134.0, 127.1, 122.6, 121.6, 119.4 (q, *J* = 274.7 Hz), 103.1, 35.2 (q, *J* = 4.6 Hz), 66.4 (2C), 45.3 (2C). ^19^F NMR (282 MHz, CDCl_3_): *δ* −68.60. HRMS (ESI) *m*/*z* [M + H]^+^ calcd for C_15_H_14_F_3_N_4_OS: 355.0835; found: 355.0832.

*5-(N-Benzylamino)-3-phenyl-7-(trifluoromethyl)pyrazolo[1,5-a]pyrimidine* (**6d**). The purification of the crude product by chromatography on silica gel is carried out using (PE/EtOAc: 9.5/0.5) to afford **6d** as a yellow solid in 80% yield. m.p. 173–175 °C; ^1^H NMR (300 MHz, CDCl_3_): *δ* 8.34 (s, 1H), 7.98 (d, *J* = 7.4 Hz, 2H), 7.46–7.25 (m, 7 H), 7.24 (t, *J* = 7.4 Hz, 1H), 6.49 (s, 1H), 5.43 (s, 1H), 4.78 (d, *J* = 5.4 Hz, 2 H); ^13^C NMR (75 MHz, CDCl_3_): *δ* 153.7, 145.4, 143.6, 137.7, 135.0 (q, *J* = 37.1 Hz), 132.2, 128.8 (2C), 128.6 (2C), 127.9 (2C), 127.9 (2C), 125.8 (2C), 119.3 (q, *J* = 274.4 Hz), 107.2, 98.5, 45.9; ^19^F NMR (282 MHz, CDCl_3_): *δ* −68.69; HRMS (ESI) *m*/*z* [M + H]^+^ calcd for C_20_H_16_F_3_N_4_: 369.1322; found: 369.1319.

*5-(N-Benzylamino)-3-(thien-2-yl)-7-(trifluoromethyl)pyrazolo[1,5-a]pyrimidine* (**6e**). The purification of the crude product by chromatography on silica gel is carried out using (PE/EtOAc: 9.7/0.3) to afford **6e** as a yellow solid in 76% yield. m.p. 156–158 °C; ^1^H NMR (300 MHz, CDCl_3_): *δ* 8.24 (s, 1H), 7.47–7.34 (m, 6H), 7.23 (d, *J* = 4.6 Hz, 1H), 7.1 (dd, *J* = 5.3, 4.0 Hz, 1H), 6.45 (s, 1H), 5.43 (br s, 1H), 4.8 (d, *J* = 5.4 Hz, 2H); ^13^C NMR (75 MHz, CDCl_3_): *δ* 153.8, 144.6, 142.9, 137.6, 135.0 (q, *J* = 37.0 Hz), 133.9, 128.9 (2C), 128.1 (2C), 127.9, 127.2, 122.7, 122.0, 119.2 (q, *J* = 274.5 Hz), 103.2, 98.7, 45.8; ^19^F NMR (282 MHz, CDCl_3_): *δ* −68.62; HRMS (ESI) *m*/*z* [M + H]^+^ calcd for C_18_H_14_F_3_N_4_S: 375.0886; found: 375.0883.

*3-([1,1’-Biphenyl]-3-yl)-5-[N-(4-methoxybenzylamino)]-7-(trifluoromethyl)pyrazolo[1,5-a]pyrimidine* (**6f**). The purification of the crude product by chromatography on silica gel is carried out using (PE/EtOAc: 9/1) to afford **6f** as a yellow solid in 90% yield. m.p. 124–126 °C; ^1^H NMR (300 MHz, CDCl_3_): *δ* 8.39 (s, 1 H), 8.36 (s, 1H), 7.99–7.96 (m, 1H), 7.69–7.67 (m, 2H), 7.50 (d, *J* = 7.6 Hz, 2H), 7.47–7.45 (m, 1H), 7.42 (d, *J* = 7.6 Hz, 2H), 7.35 (d, *J* = 8.6 Hz, 2H), 6.90 (d, *J* = 8.6 Hz, 2H), 6.48 (s, 1H), 5.40 (br s, 1H), 4.74 (d, *J* = 5.2 Hz, 2H), 3.83 (s, 3H); ^13^C NMR (75 MHz, CDCl_3_): *δ* 159.3, 153.7, 145.9, 143.6, 141.5, 141.5, 135.0 (q, *J* = 37.1 Hz), 132.8, 129.7, 129.3 (2C), 129.0, 128.7 (2C), 127.2, 127.2 (2C), 124.7, 124.5 (2C), 119.3 (q, *J* = 274.4 Hz), 114.3 (2C), 107.0, 98.6, 55.3, 45.3; ^19^F NMR (282 MHz, CDCl_3_): *δ* −68.64; HRMS (ESI) *m*/*z* [M + H]^+^ calcd for C_27_H_22_F_3_N_4_O: 475.1740; found: 475.1739

*3-[(4-Hydroxyméthyl)phenyl]-5-((4-methoxybenzyl)amino)-7-(trifluoromethyl)pyrazolo[1,5-a]pyrimidine* (**6g**). The purification of the crude product by chromatography on silica gel is carried out using (PE/EtOAc: 7.5/2.5) to afford **6g** as a yellow solid in 82% yield. m.p. 184–186 °C; ^1^H NMR (300 MHz, Acetone-*d*_6_): *δ* 8.40 (s, 1H), 8.07 (d, *J* = 8.3 Hz, 2H), 7.61 (br s, 1H), 7.44 (d, *J* = 8.7 Hz, 2H), 7.40 (d, *J* = 8.3 Hz, 2H), 6.93 (s, 1H), 6.93 (d, *J* = 8.7 Hz, 2H), 4.73 (d, *J* = 5.6 Hz, 2H), 4.65 (d, *J* = 5.5 Hz, 2H), 4.17 (t, *J* = 5.7 Hz, 1H), 3.78 (s, 3H); ^13^C NMR (75 MHz, Acetone-*d*_6_): *δ* 159.1, 154.3, 145.8, 142.6, 139.4, 133.8 (q, J = 36.8 Hz), 131.5, 130.8, 129.1 (2C), 126.9 (2C), 125.2 (2C), 119.7 (q, J = 273.3 Hz), 113.8 (2C), 106.1, 99.8, 63.8, 54.6, 44.5; ^19^F NMR (282 MHz, Acetone-*d*_6_): *δ* −69.15; HRMS (ESI) *m*/*z* [M + H]^+^ calcd for C_22_H_20_F_3_N_4_O_2_: 429.1533; found: 429.1531.

*5-[N-(4-Methoxybenzyl)amino]-3-(1H-pyrazol-4-yl)-7-(trifluoromethyl)pyrazolo[1,5-a]pyrimidine* (**6h**). The purification of the crude product by chromatography on silica gel is carried out using (PE/EtOAc: 8/2) to afford **6h** as a yellow solid in 50% yield. m.p. 197–199 °C; ^1^H NMR (300 MHz, Acetone-*d*_6_): *δ* 12.08 (s, 1H), 8.24 (s, 1 H), 8.14 (s, 2H), 7.43 (d, *J* = 8.7 Hz, 2H), 6.92 (d, *J* = 8.7 Hz, 2H), 6.88 (s, 1H), 4.74 (d, *J* = 5.5 Hz, 2H), 3.78 (s, 3H). ^13^C NMR (75 MHz, Acetone-*d*_6_): *δ* 159.1, 153.95, 145.0, 142.05, 135.55, 133.7 (q, *J* = 36.8 Hz), 130.8, 129.1 (2C), 122.0, 119.7 (q, *J* = 273.2 Hz), 113.8 (2C), 112.5, 110.1, 99.5, 54.6, 44.4. ^19^F NMR (282 MHz, Acetone-*d*_6_): *δ* −69.22. HRMS (ESI) *m*/*z* [M + H]^+^ calcd for C_18_H_16_F_3_N_6_O: 389.1332; found: 389.1331.

*5-[N-(4-Methoxybenzyl)amino]-3-(pyridin-3-yl)-7-(trifluoromethyl)pyrazolo[1,5-a]pyrimidine* (**6i**). The purification of the crude product by chromatography on silica gel is carried out using (PE/EtOAc: 8/2) to afford **6i** as a yellow solid in 91% yield. m.p. 192–194 °C; ^1^H NMR (300 MHz, CDCl_3_): *δ* 9.29 (s, 1H), 8.45 (s, 1H), 8.33 (s, 1H), 8.29 (d, *J* = 8.0 Hz, 1H), 7.39–7.37 (m, 1H), 7.33 (d, *J* = 8.5 Hz, 2H), 6.90 (d, *J* = 8.5 Hz, 2H), 6.54 (s, 1H), 5.84 (br s, 1H), 4.68 (d, *J* = 5.2 Hz, 2H), 3.80 (s, 3H). ^13^C NMR (75 MHz, CDCl_3_): *δ* 158.9, 153.7, 146.4, 145.9, 142.5, 134.5 (q, *J* = 37.1 Hz), 132.2, 129.1, 128.9 (2C), 128.4, 123.2, 118.8 (q, *J* = 274.3 Hz), 113.8 (3C), 103.3, 98.8, 54.9, 45.0. ^19^F NMR (282 MHz, CDCl_3_): *δ* −68.58. HRMS (ESI) *m*/*z* [M + H]^+^ calcd for C_20_H_17_F_3_N_5_O: 400.1380; found: 400.1376.

*3-(Dibenzo[b,d]thiophen-4-yl)-5-[N-(4-methoxybenzyl)amino]-7-(trifluoromethyl)pyrazolo[1,5-a]pyrimidine* (**6j**). The purification of the crude product by chromatography on silica gel is carried out using (PE/EtOAc: 9/1) to afford **6j** as a yellow solid in 92% yield. m.p. 198–200 °C; ^1^H NMR (300 MHz, Acetone-*d*_6_): *δ* 8.59 (s, 1H), 8.38–8.35 (m, 1H), 8.25 (dd, *J* = 7.7, 1.1 Hz, 1H), 8.23 (dd, *J* = 7.7, 1.1 Hz, 1H), 8.03–8.00 (m, 1H), 7.67 (t, *J* = 4.8 Hz, 1H), 7.63 (t, *J* = 7.7 Hz, 1H), 7.54 (dd, *J* = 5.7, 3.4 Hz, 2H), 7.41 (d, *J* = 8.7 Hz, 2H), 7.03 (s, 1H), 6.92 (d, *J* = 8.7 Hz, 2H), 4.69 (d, *J* = 5.7 Hz, 2H), 3.77 (s, 3H). ^13^C NMR (75 MHz, Acetone-*d*_6_): *δ* 159.1, 154.6, 146.5, 143.3, 139.0, 136.9, 136.15, 135.85, 133.9 (q, *J* = 36.8 Hz), 130.6, 129.2 (2C), 127.9, 126.9, 126.7, 124.9, 124.6, 122.6, 121.75, 119.7 (q, *J* = 273.2 Hz), 119.25, 113.75 (2C), 105.3, 100.5, 54.6, 44.4. ^19^F NMR (282 MHz, Acetone-*d*_6_): *δ* −69.04. HRMS (ESI) *m*/*z* [M + H]^+^ calcd for C_27_H_20_F_3_N_4_OS: 505.1304; found: 505.1303.

*5-N-[4-(Hydroxycyclohexyl)amino]-3-(naphthalen-2-yl)-7-(trifluoromethyl)pyrazolo[1,5-a]pyrimidine* (**6k**). The purification of the crude product by chromatography on silica gel is carried out using (PE/EtOAc: 9/1) to afford **6k** as a yellow solid in 91% yield. m.p. 251–253 °C; ^1^H NMR (300 MHz, Acetone-*d*_6_): *δ* 8.78 (s, 1H), 8.56 (s, 1H), 8.28 (dd, *J* = 8.6, 1.7 Hz, 1H), 7.94 (d, *J* = 8.6 Hz, 1H), 7.89 (d, *J* = 8.5 Hz, 2H), 7.53–7.41 (m, 2H), 7.22 (d, *J* = 6.7, 1H), 6.86 (s, 1H), 4.16–4.05 (m, 1H), 3.86 (d, *J* = 4.4 Hz, 1H), 3.73–3.65 (m, 1H), 2.38–2.34 (m, 2H), 2.08–2.06 (m, 2H), 1.66–1.42 (m, 4H); ^13^C NMR (75 MHz, DMSO-*d*_6_): *δ* 154.1, 146.3, 142.8, 134.1, 133.8 (q, *J* = 36.6 Hz), 131.8, 130.7, 127.8, 127.6, 127.60, 126.05, 124.9, 124.2, 123.0, 119.7 (q, *J* = 273.3 Hz), 105.6, 100.0, 69.0, 50.3, 34.1 (2C), 29.7 (2C); ^19^F NMR (282 MHz, Acetone-*d*_6_): *δ* −62.10; HRMS (ESI) *m*/*z* [M + H]^+^ calcd for C_23_H_22_F_3_N_4_O: 427.1740; found: 427.1739.

*N-[4-(Hydroxycyclohexyl)amino]-3-(3-(trifluoromethyl)phenyl)-7-(trifluoromethyl)pyrazolo[1,5-a]pyrimidine* (**6l**). The purification of the crude product by chromatography on silica gel is carried out using (PE/EtOAc: 9.2/0.8) to afford **6l** as a yellow solid in 77% yield. m.p. 257–259 °C; ^1^H NMR (300 MHz, Acetone-*d*_6_): *δ* 8.77 (s, 1H), 8.54 (s, 1H), 8.26 (d, *J* = 7.8 Hz, 1H), 7.63 (t, *J* = 7.8 Hz, 1H), 7.51 (d, *J* = 7.8 Hz, 1H), 7.27 (d, *J* = 6.9 Hz, 1H), 6.86 (s, 1H), 4.09–4.05 (m, 1H), 3.82 (d, *J* = 6.9 Hz, 1H), 3.67–3.60 (m, 1H), 2.29–2.26 (m, 2H), 2.08–2.05 (m, 2H), 1.54–1.42 (m, 4H). ^13^C NMR (75 MHz, Acetone-*d*_6_): *δ* 154. 4, 146.4, 142.5, 134.2, 133.8 (q, *J* = 36.5 Hz), 130.3 (q, *J* = 31.5 Hz), 129.2, 128.0, 124.8 (q, *J* = 272.0 Hz), 121.6 (q, *J* = 3.9 Hz), 121.2 (q, *J* = 3.7 Hz), 119.6 (q, *J* = 273.2 Hz), 104.1, 100.3, 68.9, 50.0, 33.9 (2C), 29.8 (2C). ^19^F NMR (282 MHz, Acetone-*d*_6_): *δ* −63.01, −69.09. HRMS (ESI) *m*/*z* [M + H]^+^ calcd for C_20_H_19_F_6_N_4_O: 445.1458; found: 445.1456.

*3-(Naphthalen-2-yl)-5-[2-(tert-butoxycarboxylamino)-N-ethylamino]-7-(trifluoromethyl)pyrazolo[1,5-a]pyrimidine* (**6m**). The purification of the crude product by chromatography on silica gel is carried out using (PE/EtOAc: 8.5/1.5) to afford **6m** as a yellow solid in 74% yield. m.p. 183–185 °C; ^1^H NMR (300 MHz, Acetone-*d*_6_): *δ* 8.73 (s, 1 H), 8.57 (s, 1H), 8.33 (dd, *J* = 8.6, 1.6 Hz, 1H), 7.96–7.85 (m, 3H), 7.46–7.43 (m, 3H), 6.91 (s, 1H), 6.32 (br s, 1H), 3.82 (q, *J* = 5.9 Hz, 2H), 3.55 (q, *J* = 6.0 Hz, 2H), 1.40 (s, 9H); ^13^C NMR (75 MHz, Acetone-*d*_6_): *δ* 156.2, 155.0, 146.1, 142.9, 134.1, 131.8, 130.5, 127.9 (2C), 127.5, 125.9, 124.9, 124.4, 123.1, 119.7, 119.7 (q, *J* = 273.3 Hz), 105.9, 100.0, 78.0, 41.6, 39.4, 27.7 (3C); ^19^F NMR (282 MHz, Acetone-*d*_6_): *δ* −69.06; HRMS (ESI) *m*/*z* [M + H]^+^ calcd for C_24_H_25_F_3_N_5_O_2_: 472.1955; found: 472.1953.

### 3.5. Deprotection of N-Boc Protected Amine by TFA: Synthesis of 5-[2-Amino-N-ethylamino]-3-(naphthalen-2-yl)-7-(trifluoromethyl)pyrazolo[1,5-a]pyrimidine (***9***)

A solution of the *N*-Boc protected amine **6m** in 5 mL of trifluoroacetic acid was stirred under argon atmosphere at room temperature for 6 h. After the trifluoacetic acid was removed under vacuum, the residue was mixed with water (1 mL), and the mixture was treated with an ammonia solution (10%) to pH = 8. The formed solid was filtered and washed with water and Et_2_O to give product **9** as a yellow solid in 95% yield. m.p. 171–173 °C; ^1^H NMR (300 MHz, Acetone-*d*_6_): *δ* 8.57 (s, 1H), 8.45 (s, 1H), 8.19 (dd, *J* = 8.6, 1.7 Hz, 1H), 7.83 (d, *J* = 8.5 Hz, 1H), 7.82 (d, *J* = 8.0 Hz, 1H), 7.77 (d, *J* = 8.0 Hz, 1H), 7.38 (td, *J* = 7.5, 1.3 Hz, 1H), 7.33 (td, *J* = 7.5, 1.3 Hz, 1H), 6.83 (s, 1H), 4.18–4.20 (m, 2H), 4.05–4.03 (m, 2H). HRMS (ESI) *m*/*z* [M + H]^+^ calcd for C_24_H_25_F_3_N_5_O_2_: 372.1455; found: 372.1453.

## 4. Conclusions

In conclusion, an efficient strategy to synthesize 3,5-disubstituted pyrazolo[1,5-*a*]pyrimidines bearing CF_3_ group at C-7 was developed, starting from the building block, 3-bromo-7-(trifluoromethyl)pyrazolo[1,5-*a*]pyrimidin-5-one. A sequential S*_N_*Ar with various amines and Suzuki cross-coupling were achieved at C-5 and C-3, respectively. The activation of the C-O bond of the amide function followed by the addition of several amines or *p*-methoxythiophenol allowed the functionalization of pyrazolo[1,5-*a*]pyrimidine core at C-5. The arylation at C-3 was carried out in good to excellent yields under Suzuki cross-coupling, requiring XPhosPdG2/XPhos as a catalytic system and potassium carbonate as a base in a mixture of EtOH/H_2_O (4/1) at 135 °C under microwave irradiation. Further studies are underway in our laboratory following this strategy in order to develop highly biologically active pyrazolo[1,5-*a*]pyrimidines.

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
