# Peer review of "Efficient Access to 3,5-Disubstituted 7-(Trifluoromethyl)pyrazolo[1,5-*a*]pyrimidines Involving S*_N_*Ar and Suzuki Cross-Coupling Reactions"

_molecules, 2020, doi:10.3390/molecules25092062_

Round 1
Reviewer 1 Report
Abarbri and co-workers describe the C3- functionalization of 7-CF3-pyrazolo-pyrimidines using a Suzuki cross-coupling reaction. The present study is a spin-off of a previously published paper (ref 41, Synthesis 2018 1675). Very related reactions on pyrazolo-pyrimidines (ACS Med ChemLett 2016-671; BioorgMedChem 2015-7240 + many patents) have been previously reported in the literature. Moreover, although these compounds could be potentially interesting, bio-activities are not disclosed ? At this stage, publication is not recommended.Author Response
Comments and Suggestions for Authors:
Abarbri and co-workers describe the C3- functionalization of 7-CF3-pyrazolo-pyrimidines using a Suzuki cross-coupling reaction. The present study is a spin-off of a previously published paper (ref 41, Synthesis 2018 1675). Very related reactions on pyrazolo-pyrimidines (ACS Med ChemLett 2016-671; BioorgMedChem 2015-7240 + many patents) have been previously reported in the literature. Moreover, although these compounds could be potentially interesting, bio-activities are not disclosed ? At this stage, publication is not recommended.
Response: I disagree with referee 1 because our approach to prepare 3,5-Disubstituted 7-(Trifluoromethyl)pyrazolo[1,5-a]pyrimidines is new, completely different and have more advantages compared to related reactions previously reported on pyrazolo-pyrimidines (ACS Med ChemLett 2016-671; BioorgMedChem 2015-7240 + many patents)
By the way the bio-activities of the final C3-functionalization of 7-CF3-pyrazolo-pyrimidines : the biological tests are underway in collaboration with an expert laboratory and all the results including the study of the structure-activity relationship will be published in a journal specialized in medicinal chemistry.
Reviewer 2 Report
This paper describes the synthesis of 3,5-disubstituted 7-(trifluoromethyl)pyrazolo[1,5-a]pyrimidines.
Starting from 3-aminopyrazole, 3-bromo-7-(trifluoromethyl)pyrazolo[1,5-a]pyrimidine-5-one was prepared by their own reported method. By using PyBroP, one pot SNAr reaction of the above compound gave the 5-amino derivatives in good yield. Subsequently, the bromide was allowed to react with boronic acid under the condition of Suzuki-Miyaura coupling condition. The authors successfully obtained the desired product in good yield. The paper may be worthy of being published in Molecules after some modifications. The following points may be considered.
Many abbreviations were used in the manuscript. The list of them would be helpful for the broad readership.
The authors found that XPhosPdG2 gave superior results in the Suzuki-Miyaura coupling step. If the reason for this improvement is available, some comment on the difference should be mentioned. There are so many Pd catalysts have been developed, the possibility of some other Pd catalysts should also be tested.
In this step, the reaction was performed at 135 °C in ethanol/water. The authors should explain how this temperature was maintain during the reaction. In Table 1, isolated yield is only shown for the entry 6. Yield for other entries should be included.
Author Response
Comments and Suggestions for Authors:
Many abbreviations were used in the manuscript. The list of them would be helpful for the broad readership
Response: Most of the abbreviations used in the manuscript are known and widely used in organic chemistry. However, for the most complex catalysts and ligands, we have added below the table 1 the nomenclature of their abbreviations.
Comments and Suggestions for Authors:
The authors found that XPhosPdG2 gave superior results in the Suzuki-Miyaura coupling step. If the reason for this improvement is available, some comment on the difference should be mentioned.
Response:
The nature of the palladium precatalyst or source of Palladium is important for efficient cross-coupling because it determines the formation and availability of active LnPd (0) species. If the active ligated Pd (0) is formed easily and under mild conditions, it can reduce the undesirable side reaction of the two coupling partners, such as the dehalogenation of aryl halides for example. This is what we observed in our case using the catalytic system (XPhosPdG2/XPhos). Since it is the only catalyst among all the other catalysts tested which allowed us to avoid the debromination reaction. The same results were observed by Cankar et al. in the case of Suzuki-Miyaura coupling reaction of halogenated aminopyrazoles (J. Org. Chem. 2017, 82, 157-169 and Eur. J. Org. Chem. 2016, 2013-2023).
Comments and Suggestions for Authors:
There are so many Pd catalysts have been developed, the possibility of some other Pd catalysts should also be tested
Response: Our main objective was to carry out the Suzuki cross-coupling reaction in excellent yield without the formation of side reaction like the debromination. Since we found the good catalytic system (XPhosPdG2/XPhos) that allowed us to avoid the formation of the debrominated product, it seemed useless to try other catalysts.
Comments and Suggestions for Authors:
In this step, the reaction was performed at 135 °C in ethanol/water. The authors should explain how this temperature was maintain during the reaction.
Response: The temperatures were measured on the surface of the vial with an IR sensor (measuring range: 30 to 300 °C; uncertainty: ±5 °C) and were measured with high precision in the reaction mixture with a ruby thermometer (measuring range: 30 to 300 °C; uncertainty: ±2 °C) that could be read directly from the instrument screen (this detail was mentioned in experimental part).
Comments and Suggestions for Authors:
In Table 1, isolated yield is only shown for the entry 6. Yield for other entries should be included.
Response: We did not determine the isolated yields in the other entries (1-6 and 7-8), simply because in these cases there was the formation of the debrominated product 7, and for example, the best conversion (entry 6) did not exceed 84%, indicating that the isolated yield would be lower than this value. Moreover, the separation of the mixture of the two products (expected product 6a and the debromination product 7) has proved to be a difficult operation since the two products 6a and 7 have almost the same Rf on silica whatever the eluent used.
Reviewer 3 Report
The paper by Abarbri and coworkers reported the synthesis of 3,5-Disubstituted-7-(Trifluoromethyl)pyrazolo[1,5-a]pyrimidines4 using SNAr and Suzuki Cross-Coupling reactions. The reported paper is the development and the further exploration of the papers that have been reported over the years by the same group and referenced in this paper from 38 to 41. The paper has used SNAr and Suzuki coupling reactions which have been demonstrated in a similar system multiple times. I strongly believe the paper lacks novelty as the synthesis is the extension of what they have done over the years. The first step to get fluorinated pyrazolo[1,5-a]pyrimidin-5-one and the SNAr reaction in almost same system has reported before by the same group (ref. 41). The only thing difference between the previous paper from the reported paper is the selective bromination to get 3-bromo-7-(trifluoromethyl)pyrazolo[1,5-a]pyrimidin-5-one which has been demonstrated before by various groups and even patented. The author has used this bromine for further functionalization. Although the paper utilized the previously known methods, the complexity and the beautiful design of the molecule cannot be denied. Furthermore, the authors have designed the analogues of the compounds which have a nanomolar activity against Pim1 kinase. I recommend publication of this article in Molecules, but after some revisions below to improve the manuscript.
- Is it possible to run the reaction in lower ligand and catalyst loadings?
- Why 10 mol% of XPhos is required as the catalyst contains XPhos in it?
- Why yield increases going from Na2CO3 to K2CO3? Is it because of the size difference? If yes, then what is the tentative explanation of why the size of cation matters?
- Does polar aprotic solvent works or polar protic solvents is required?
Author Response
Comments and Suggestions for Authors:
Is it possible to run the reaction in lower ligand and catalyst loadings?
Response: It is entirely possible to carry out the coupling reaction with a total conversion rate using 5 mol% of catalyst and 5 mol% of ligand, but the yield of the coupling product 6c does not exceed 85%. An additional entry has been added in the Table 1 for this case as well as a sentence in the text.
Comments and Suggestions for Authors:
Why 10 mol% of XPhos is required as the catalyst contains XPhos in it?
Response: PdXPhosG2 is a palladium(II) and it’s required to add XPhos as a ligand to form in situ a palladium(0) which is necessary to make an oxidative addition of carbon-bromide bond.
Comments and Suggestions for Authors:
Why yield increases going from Na2CO3 to K2CO3? Is it because of the size difference? If yes, then what is the tentative explanation of why the size of cation matters?
Response:
This difference in behavior between K2CO3 and Na2CO3 could be linked to the size of the solvated cation. Sodium is more solvated by water molecules than potassium owing to the fact that polarizing effect of sodium (charge/rasius) is higher. Hence the interaction between the cation and the negative charge of the carbonate ion is lower when sodium is used and hence the basicity of Na2CO3 is higher than that of K2CO3 in water.
Comments and Suggestions for Authors:
Does polar aprotic solvent works or polar protic solvents is required?
Response: In Suzuki cross-coupling reaction, polar aprotic solvents work also but polar protic solvents work better and give good yields because they dissolve better the base which is necessary to form the ate complex with boronic acid and allowed the transmetallation with palladium.
Reviewer 4 Report
Abarbri and co-workers describe a protocol with successive synthesis to give access to 3,5-disubstituted-3,7-(trifluoromethyl)pyrazolo[1,5-a]pyrimidines involving SNAr and Suzuki Cross-Coupling Reactions. The initial protocol seems similar to their previous reported manuscript, but the final products are new and interesting. It would be nice to know also the biological activity of their own compounds, beyond to provide the values of IC50 reported by other authors. But I suppose that the authors are not expert in biological properties, otherwise they should perform these additional proofs.
However, I would like to see and to check the spectra of all new compounds; otherwise I will not accept the manuscript. The author must provide the supplementary information.
Author Response
Comments and Suggestions for Authors:
It would be nice to know also the biological activity of their own compounds, beyond to provide the values of IC50 reported by other authors. But I suppose that the authors are not expert in biological properties, otherwise they should perform these additional proofs.
Response:
The reviewer is right. We are not experts in biological properties. However, activity tests will be carried out by another laboratory specialized in the field and with that we collaborate. The results obtained will be published in a journal specialized in medicinal chemistry.
Comments and Suggestions for Authors:
However, I would like to see and to check the spectra of all new compounds; otherwise I will not accept the manuscript. The author must provide the supplementary information
Response: All 1H NMR, 13C NMR and 19F NMR spectra of new products were sent in a Pdf file of supplementary information, with the exception of product 9 where only the 1H NMR spectrum was transmitted because we were not able to perform the 13C NMR and 19F NMR analysis due to the closure of our laboratory due to a cofid-19 problem. We are of course committed to sending you the missing analyzes as soon as we resume our research activity. We can also postpone these analyzes in the next manuscript dealing with the biological activity of these compounds. All HRMS analyzes of new products have been reported in the manuscript.
Round 2
Reviewer 1 Report
I didn't change my mind about this article and leave the final decision to the editorial office. This paper, as a spin-off of ref 41, appears as a very small contribution to warrant publication in Molecules.
- The answer ‘our approach is new, completely different and have more advantages compared‘ is not argued at all.
- Biological activities are not described: “structure-activity relationship will be published in a journal specialized in medicinal chemistry” ... As SAR comes with synthesis, I suggest publication as a full paper including synthesis and SAR.
- The suggested ref ‘BioorgMedChem 2015-7240’ has not even be added in the list of references….
Reviewer 4 Report
I accept the manuscript in present form